# Association of the *SH2B1* rs7359397 Gene Polymorphism with Steatosis Severity in Subjects with Obesity and Non-Alcoholic Fatty Liver Disease

**DOI:** 10.3390/nu12051260

**Published:** 2020-04-29

**Authors:** Nuria Perez-Diaz-del-Campo, Itziar Abete, Irene Cantero, Bertha Araceli Marin-Alejandre, J. Ignacio Monreal, Mariana Elorz, José Ignacio Herrero, Alberto Benito-Boillos, Jose I. Riezu-Boj, Fermín I. Milagro, Josep A. Tur, J. Alfredo Martinez, M. Angeles Zulet

**Affiliations:** 1Department of Nutrition, Food Science and Physiology, Faculty of Pharmacy and Nutrition, University of Navarra, 31008 Pamplona, Spain; nperezdiaz@alumni.unav.es (N.P-D.-d.-C.); icgonzalez@unav.es (I.C.); bmarin.1@alumni.unav.es (B.A.M.-A.); jiriezu@unav.es (J.I.R.-B.); fmilagro@unav.es (F.I.M.); jalfmtz@unav.es (J.A.M.); 2Centre for Nutrition Research, Faculty of Pharmacy and Nutrition, University of Navarra, 31008 Pamplona, Spain; 3Biomedical Research Centre Network in Physiopathology of Obesity and Nutrition (CIBERobn), Instituto de Salud Carlos III, 28029 Madrid, Spain; pep.tur@uib.es; 4Navarra Institute for Health Research (IdiSNA), 31008 Pamplona, Spain; jimonreal@unav.es (J.I.M.); marelorz@unav.es (M.E.); iherrero@unav.es (J.I.H.); albenitob@unav.es (A.B.-B.); 5Clinical Chemistry Department, Clínica Universidad de Navarra, 31008 Pamplona, Spain; 6Department of Radiology, Clínica Universidad de Navarra, 31008 Pamplona, Spain; 7Liver Unit, Clinica Universidad de Navarra, 31008 Pamplona, Spain; 8Centro de Investigación Biomédica en Red de Enfermedades Hepáticas y Digestivas (CIBERehd), 28029 Madrid, Spain; 9Research Group on Community Nutrition and Oxidative Stress, University of Balearic Islands & Balearic Islands Institute for Health Research (IDISBA), 07122 Palma, Spain

**Keywords:** NAFLD, obesity, steatosis, *SH2B1*, polymorphisms

## Abstract

Non-alcoholic fatty liver disease (NAFLD) is a major cause of liver disease worldwide. Some genetic variants might be involved in the progression of this disease. The study hypothesized that individuals with the rs7359397 T allele have a higher risk of developing severe stages of NAFLD compared with non-carriers where dietary intake according to genotypes could have a key role on the pathogenesis of the disease. *SH2B1* genetic variant was genotyped in 110 overweight/obese subjects with NAFLD. Imaging techniques, lipidomic analysis and blood liver biomarkers were performed. Body composition, general biochemical and dietary variables were also determined. The *SH2B1* risk genotype was associated with higher HOMA-IR *p* = 0.001; and Fatty Liver Index (FLI) *p* = 0.032. Higher protein consumption (*p* = 0.028), less mono-unsaturated fatty acid and fiber intake (*p* = 0.045 and *p* = 0.049, respectively), was also referred to in risk allele genotype. Lipidomic analysis showed that T allele carriers presented a higher frequency of non-alcoholic steatohepatitis (NASH) (69.1% vs. 44.4%; *p* = 0.006). In the genotype risk group, adjusted logistic regression models indicated a higher risk of developing an advanced stage of NAFLD measured by FLI (OR 2.91) and ultrasonography (OR 4.15). Multinomial logistic regression models showed that risk allele carriers had higher liver fat accumulation risk (RRR 3.93) and an increased risk of NASH (RRR 7.88). Consequently, subjects carrying the T allele were associated with a higher risk of developing a severe stage of NAFLD. These results support the importance of considering genetic predisposition in combination with a healthy dietary pattern in the personalized evaluation and management of NAFLD.

## 1. Introduction

Non-alcoholic fatty liver disease (NAFLD) is a frequent hepatic manifestation of metabolic syndrome with an estimated prevalence of 20–30% in the general population, whose rates rise with the increasing incidence of obesity [1,2]. NAFLD is described as an excessive hepatic fat deposition in the absence of history of alcohol abuse or other causes of secondary hepatic steatosis [3,4]. This disease encompasses a spectrum of clinical conditions, which can range from simple fat accumulation to non-alcoholic steatohepatitis (NASH) to advanced fibrosis leading to cirrhosis or hepatocellular carcinoma (HCC) and death [5,6]. The pathogenesis of NAFLD is multifactorial [7,8]. A sedentary lifestyle, obesity and related comorbidities such as diabetes, dyslipidemias, insulin resistance and other metabolic syndrome components are important risk factors associated with the development of NAFLD [9,10]. Besides, the current treatment of NAFLD is based on lifestyle interventions, such as changes in dietary patterns [6]. Thus, weight loss, exercise and healthy eating habits such as a Mediterranean lifestyle have been proposed as the main strategies in the reduction of NAFLD-associated comorbidities and to improve quality of life [2,3].

Not all subjects with similar sociodemographic and physical characteristics develop NAFLD and not all subjects with NAFLD develop more advanced stages of the disease, suggesting that important inter-individual differences concerning the mechanisms are involved in the pathogenesis and progression of NAFLD [11,12,13,14].

Research in this area has revealed that NAFLD development and progression towards more advanced stages have a genetic component [15,16,17]. The identification of genes that might confer a higher risk for the development of severe NAFLD as well as metabolic alterations directly related to the disease could be of special interest [1,17]. In order to improve and individualize the treatment, there are some identified genetic variants associated with increased liver fat accumulation, high risk for NAFLD and HCC development [18,19]. Advances in nutritional research and genomics may help to understand and improve the personalization of NAFLD treatment, taking into account genetic and gene–nutrient interactions [6,13]. Thus, there have been reported differences in cholesterol, adiposity and insulin resistance outcomes according to obesity-related variants in response to dietary interventions [20]. Moreover, the combination of environmental data and genetics was showed as an important predictor of blood lipid phenotypes [21].

Genetic variants more closely related to obesity might also be linked to NAFLD such as the Src homology (*SH2B*), among others. This gene family contains three members of adaptor proteins *(SH2B1*, *2* and *3*), being highly expressed in the liver [22]. Its potential mechanism may be as an adaptor protein, wherein *SH2B1* is implicated in several transduction processes such as enhancing JACK2 or the PI3-kinase pathway [23,24]. A genetic disruption of this gene has been associated with severe leptin resistance, energy imbalance, obesity and type 2 diabetes in humans, which are common comorbidities related to NAFLD [24]. In addition, it has been recently published that *SH2B1* can regulate the migration, proliferation and differentiation of cells, which could influence the development of some cancers [23]. In animal studies, an association has been drawn between *SH2B1* with an increasing hepatic lipid content and/or VLDL secretion, promoting hepatic steatosis in mice [22]. In humans, studies of this genetic variant linked to NAFLD have not been reported to date. However, among the *SH2B1* genetic variants identified as related to obesity traits, the polymorphism rs7359397 has been associated with glycosylated hemoglobin [25] and insulin sensitivity [24,25]. 

In this context, the aim was to analyze the effect of the *SH2B1* genetic variant related to NAFLD as well as possible associations between this polymorphism and diet, in order to identify possible gene–diet interactions that help clarify the role of this specific polymorphism in the pathogenesis of non-alcoholic fatty liver disease. Therefore, the study hypothesized that individuals with the rs7359397 T allele have a higher risk of developing severe stages of NAFLD compared with non-carriers where dietary intake according to genotypes could have a key role on the pathogenesis of the disease.

## 2. Materials and Methods

### 2.1. Study Population

The current study encompassed 127 men and women, overweight or obese (BMI ≥ 27.5 and < 40 kg/m^2^) with ultrasound-confirmed liver steatosis following accepted clinical criteria, as previously reported [10]. The analyses were conducted within the FLiO project (Fatty Liver in Obesity), a randomized controlled trial (www.clinicaltrials.gov; NCT03183193), which follows the Consort 2010 guidelines. The study was approved by the Ethics Committee of the University of Navarra (54/2015). All participants gave written informed consent for their participation in accordance with the Declaration of Helsinki. The exclusion criteria were endocrine disorders, (hyperthyroidism or uncontrolled hypothyroidism), known liver disease (other than NAFLD), alcohol abuse (>21 and >14 units of alcohol per week in men and women, respectively) and pharmacological treatments and a weight loss of ≥3 kg in the last 3 months, among others [14].

### 2.2. General Measurements

Anthropometric measurements (body weight, height and waist circumference) were assessed in fasting conditions following previously described standardized procedures [26]. The BMI was calculated as body weight divided by squared height (kg/m^2^). The body composition was analyzed by dual-energy X-ray absorptiometry (DXA) according to the instructions of the manufacturer (Lunar iDXA, encore 14.5, Madison, WI, USA) [27].

Blood glucose, glycosylated hemoglobin (HbA1c), homocysteine, triglycerides (TG), total cholesterol (TC), high-density lipoprotein (HDL-c), low density lipoprotein cholesterol (LDL-c), alanine aminotransferase (ALT), aspartate aminotransferase (AST) and gamma-glutamyltransferase (GGT) were measured on a suitable autoanalyzer with routine validated procedures. On the other hand, insulin, C-reactive protein (CRP) and plasma concentrations of fibroblast growth factor 21 (FGF-21) values were quantified with specific ELISA kits (Demeditec, Kiel-Wellsee, Germany) in a Triturus auto-analyzer (Grifols, Barcelona, Spain), as described by the manufacturer. The Homeostatic Model Assessment for Insulin Resistance (HOMA-IR), the TyG index (Ln[triglycerides (mg/dL)*glucose(mg/dL)]) and the Atherogenic Index of Plasma (log[triglycerides(mg/dL)/HDL-c(mg/dL)]) were also calculated as described elsewhere [4,10,26]. Physical activity was classified in 4 different categories depending on the level (sedentary, mild, moderated or elevated). Fatty Liver Index (FLI), which has been validated in a large group of subjects with or without liver disease, was also assessed. It is based on an algorithm including BMI, waist circumference, triglycerides and GGT. Accuracy was assessed by calculating the area (AUC) under the receiver operating curve (ROC) model of 0.84 with 95% confidence intervals (95% CI 0.810–0.87) in detecting fatty liver [12,28]. An index <30 points indicates the absence of fatty liver (negative likelihood ratio = 0.2) and ≥60 is a marker of fatty liver (positive likelihood ratio = 4.3) [28].

### 2.3. Dietary Assessment

The diet of the participants was assessed at baseline with a validated semiquantitative food frequency questionnaire (FFQ) of 137 items [9,11]. Each item in the questionnaire included a typical portion size. The nutrient composition of the food items was derived from accepted Spanish food composition tables. The adherence to the MedDiet was assessed with a 17-point screening questionnaire, with a final score ranging from 0 to 17 and a higher score indicating a better adherence to the MedDiet.

Glycemic Index values for single food items on the food frequency questionnaire were derived from the International Tables of Glycemic Index and Glycemic Load Values, as previously reported [12]. Total dietary Glycemic Index was estimated by multiplying the amount of available carbohydrate (g) of each food item by its GI. The sum of these products was divided by the total carbohydrate intake. The amount of carbohydrate can vary in an overall diet and because of this the concept of Glycemic Load was also applied [12].

### 2.4. Hepatic Imaging Tests

The ultrasonography methodology consisted in the evaluation of the steatosis status by visual quality of the liver echogenicity, measurements of the difference between the kidneys and the liver in the amplitude of the echo and the determination of the clarity of the structures of the blood vessels in the liver. The clinical classification was defined using a 4-point scale: normal (less than 5%), mild steatosis (5–33%), moderate steatosis (33–66%) and severe steatosis (greater than 66%), as described elsewhere [12,29,30]. Transient elastography was performed through FibroScan^®^ (Echosens, Paris, France), with the subject in the supine position and the right arm in maximum abduction. Depending on the obesity status, M and XL probes were selected under the professional criteria. Repeated shots were performed until obtaining 10 valid values of which the median was the selected value. Hepatic fibrosis and cirrhosis were considered if the stiffness median > 7 kPa or > 12 kPa, respectively [12]. Magnetic Resonance Imaging (MRI) was performed through Siemens Aera 1,5 T and it was used to detect the volume, fat and iron content of the liver (Dixon technique) as reported by the manufactures [10].

### 2.5. Metabolomics

The OWLiver^®^ test (One Way Liver S. L. Bilbao, Spain) is a non-invasive and validated lipidomic serum able to distinguish between a normal liver, simple steatosis or NASH with high accuracy [31]. The metabolomic probe used was a fasting blood probe that measures a panel of biomarkers that belong to the family of triacylglycerols, which are a reflection of the amount of fat and inflammation of the liver and, therefore, a measure of disease development of NAFLD [12]. The relative metabolite concentrations are analyzed together in an algorithm that generates the final OWLiver^®^ score, which discriminates between the three categories (No NAFLD, hepatic steatosis or NASH).

### 2.6. Genotyping

Genotype screen followed validated procedures [32,33]. A total of 110 epithelial buccal cells sweeps from participants were collected using a liquid-based kit (ORAcollect-DNA, OCR-100, DNA Genotek, Ottawa, ON, Canada). Genomic DNA was isolated using a Maxwell 16 Buccal Swab LEV DNA Purification Kit in the Maxwell 16 instrument (Promega, Madison, WI, USA) according to the instructions of the manufacturer. Genotyping of the *SH2B1* rs7359397 variant was performed by targeted next-generation sequencing using a pre-designed SNP panel (Ion AmpliSeq Custom NGS DNA Panels, Thermo Fisher Scientific Inc., Waltham, MA, USA), as previously described elsewhere [32]. Variants were identified with the Torrent Variant Caller 5.0 (Thermo Fisher Scientific) with a minimum coverage value of 20.

### 2.7. Statistical Analyses

The sample size for the main study (the FLiO study) was calculated based on the current recommendations of the AASLD on body weight to ameliorate NAFLD features [2]. Therefore, it was estimated to detect a difference of 1.0 ± 1.5 kg in body weight loss between dietary groups [10]. Furthermore, a previous study [12], where NAFLD participants were categorized considering liver fat content (MRI < 5% vs. ≥ 5%), enabled an “a posteriori” estimation, which revealed that to detect a difference of 3 ± 5 points on hepatic fat content between genotypes, with a 95% confidence interval (α = 0.05) and a statistical power of 80% (β = 0.80), the estimated sample size was n = 44 for each group. Continuous variables are expressed as means and standard deviation (SD) or as medians and interquartile ranges depending on its distribution, while qualitative categorical variables were analyzed with the chi-squared test and reported as absolute (n) and relative frequencies (%). Chi-squared test was also used to assess the Hardy–Weinberg equilibrium (HWE) concerning the alleles of risk. Distribution of variables was assessed through the Shapiro Wilk and Kolmogorov–Smirnov test. Data normality and outliers were also checked using boxplots. Those variables following a normal distribution were analyzed using parametric statistical tests while for those variables with a non-normal distribution, non-parametric statistics were applied. Descriptive statistics were used to compare baseline data of participants. For continuous variables, Student’s *t*-tests (for parametric) of independent samples and Mann-Whitney U tests (for non-parametric) were applied.

The risk of developing severe stages of NAFLD was examined by categorizing the steatosis degree in two groups (mild steatosis vs. moderate steatosis plus severe steatosis), FLI according to the median (<80 vs. ≥80) and liver fat accumulation by MRI was categorized in tertiles. Logistic regression models were set up to evaluate the association of the steatosis degree (dependent variable) with *SH2B1* genetic variant (independent variable) and to assess the influence of the genetic variant on Fatty Liver Index (FLI). Data were expressed in Odds Ratio (OR) and confidence interval. A multinomial logistic regression analysis was also performed to assess the influence of *SH2B1* on the risk of liver fat accumulation (by MRI) and to assess the risk of developing advanced stages of NAFLD (by OWL^®^) such as NASH. Data were expressed in Relative Risk Ratio (RRR) and confidence interval. Regression models were adjusted for potential confounders, some of them linked to NAFLD such as age, sex and ALT concentrations and others related to lifestyle such as adherence to MedDiet, total energy intake and physical activity. Body Mass Index was used as a covariate in the logistic regression model between steatosis degree and *SH2B1* genetic variant and in the multinomial logistic regression analysis between liver fat content (by MRI) and the polymorphism. In the FLI and OWLiver^®^-test models, BMI was not included as a covariate due to the possibility of over fitting models, since both the FLI and OWLiver-test include the BMI in its calculation.

Analyses were performed using STATA 12.0 software (Stata Corp College Station, TX, USA). All calculated *p*-values were two-tailed. Values of *p* < 0.05 were considered to be statistically significant in the analyses. 

## 3. Results

### 3.1. Characteristics of the Participants

A total of 110 participants with NAFLD were included in the study. The polymorphism was in Hardy–Weinberg Equilibrium (*p* > 0.05). The risk allele frequency (T allele) of the rs7359397_*SH2B1* genetic variant was present in about 51% of participants. No-risk genotype (CC) was present in 54 participants, while the heterozygous genotype (CT) was present in 46 subjects and the risk genotype (TT) in just 10 subjects. Because of the small sample of homozygotes for the risk allele, the sample was distributed and analyzed in two different groups: no-risk genotype (n = 54) and risk genotype, (including CT and TT subjects, n = 56). Baseline characteristics of the participants included in the present study were analyzed according to the *SH2B1* rs7359397 genetic variant. Main body composition and biochemical features are reported in Table 1. No significant differences were observed between groups in body composition variables. A marginal significant difference was observed in fat-free mass. However, when the analysis was repeated considering men and women separately, no differences were observed between carriers and non-carriers. Variables according to biochemical parameters showed higher insulin (*p* = 0.002) and lower HDL-c (*p* = 0.003) concentrations in the risk genotype group as compared to the non-risk. Likewise, risk allele carriers showed higher levels of HOMA-IR; Triglycerides/HDL-c ratio, waist*TyG index and atherogenic index (*p* = 0.001; *p* = 0.021; *p* = 0.030 and *p* = 0.012, respectively).

Concerning dietary intake and lifestyle factors (Table 2), no significant difference was observed in total energy consumption (*p* = 0.101). Regarding macronutrient distribution, significant differences were found neither in carbohydrates (*p* = 0.612) nor in lipids (*p* = 0.308); but protein percentage was significantly different between genotypes (*p* = 0.028). In addition, the total ingestion of monounsaturated fatty acids (*p* = 0.045) and fiber (*p* = 0.049) were higher in the no-risk genotype. Finally, the adherence to the Mediterranean Diet (*p* = 0.678) and physical activity (*p* = 0.685) were not different between groups.

### 3.2. Hepatic Status According to Genetic Variant Alleles

Liver blood biomarkers results (Table 3) showed no significant differences between *SH2B1* alleles in most of the variables. Risk allele carriers showed a higher Fatty Liver Index, as well as steatosis degree assessed by ultrasonography. Specifically, significant differences were observed when comparing first vs second and third steatosis degree (*p* < 0.001 and *p* = 0.049, respectively). Risk genotype also showed a higher liver fat content by MRI (*p* = 0.055). The lipidomic analysis showed that some participants, in spite of having a positive result of hepatic steatosis by the ultrasonography, registered a negative result indicating no hepatic steatosis. The frequency of participants with no NAFLD was higher in no-risk genotypes (29.6%). On the other hand, the frequency of NASH was 69.1% in risk allele carriers compared with 44.4% in non-carriers. Concretely, significant differences were found between groups when comparing No NAFLD (42.55%) vs Hepatic Steatosis (57.45%) and No NAFLD (24.39%) vs. NASH (75.61%) (*p* = 0.047 and *p* = 0.001, respectively).

### 3.3. Association between Genotype and Advanced Stages of the Disease

An adjusted logistic regression analysis was performed considering the median of the FLI (Appendix A). Results showed that those subjects carrying the T allele presented a major risk (OR 2.91) of developing a higher punctuation in FLI than homozygous subjects for the C allele (Figure 1).

When the steatosis degree was assessed by ultrasonography (Appendix A), subjects carrying the CT/TT genotype showed a significant association with a higher steatosis degree than non-carriers (*p* = 0.004). The genetic variant remained significant after adjusting for potential confounders such as sex, age and physical activity, indicating that those subjects carrying the risk allele had a 4.15 value risk of having a higher steatosis degree than non-carriers (Figure 2).

Multinomial logistic regression analyses were also performed to assess the influence of the genetic variant on liver fat accumulation by MRI as well as on the risk of developing NASH by the lipidomic test (Appendix A). The main results evidenced that risk allele carriers had an increased risk for liver fat accumulation (RRR 3.93) as well as for the development of NASH (RRR 7.88) in comparison with homozygous subjects (Figure 3 and Figure 4).

## 4. Discussion

This research project aimed to analyze the influence of a metabolism-related polymorphism on the development of advanced stages of the disease, as well as the influence of the diet in subjects with NAFLD. Interestingly, the results yielded evidence that the *SH2B1* has an impact on the development and progression of this hepatic disease. Risk allele carriers showed high risk for higher liver fat accumulation as assessed by FLI, ultrasonography and MRI and higher risk for developing NASH as assessed by metabolomics in this cross-sectional analysis.

It should be taken into account that heritability is involved in the development of advanced phases, and NAFLD has a genetic component [34,35]. Some genetic variants in genes like the PNPLA3 I148M, the TM6SF2 and the MBOAT7, have been strongly associated with the development of NAFLD-HCC [18,36,37] and with hepatic fat accumulation [16]. However, the aforementioned variants are not separated enough themselves to identify patients at risk of developing severe stadiums, most likely due to the influence and interaction of other genetic and non-genetic factors [6,18,24].

Some investigations have demonstrated that patients with advanced stages of NAFLD also present a high frequency of less prevalent gene variants [18]. Because of this, a sequence variant located in the intergenic region between genes, *rs7359397 SH2B1,* was studied. This genetic variant, which is a member of the *SH2B* family of adapter proteins that also include *SH2B2* and *SH2B3* has been reported to be pathogenic for obesity [38,39,40]. *SH2B1* and *SH2B2* are abundantly expressed in the brain, liver, heart, skeletal muscle and adipose tissue [22]. By contrast, *SH2B3* expression is restricted to hematopoietic tissue [41]. Besides this, in another genome-wide association study (GWAS), the gene *SH2B1* was suggested as a physiological enhancer of insulin receptors and downstream signaling [42]. Being pathogenic for both obesity and insulin resistance, *SH2B1* is a strong candidate for involvement in NAFLD risk and severity. Because of this, we tested the hypothesis that individuals with the rs7359397 T allele have a higher risk of developing severe stages compared with the no-risk genotype, as well as the role of diet combined with genetics in the pathogenesis of the disease.

The *SH2B1* genetic variant has also been related to body composition [15]. In the study, we found a marginal association with DXA lean mass and the T allele (*p* = 0.059), which could be related to the differences observed in sex (*p* = 0.037). Due to the possible influence of sex in body composition and biochemical variables, such as HDL-c or waist circumference, an analysis stratifying by sex was performed and no statistical differences were found. Moreover, the risk genotype was correlated with a higher value of insulin, HOMA-IR, TyG-index and higher ratio of Triglycerides/HDL-c and Waist*TyG-index, all of them related to insulin resistance. In this sense, scientific evidence revealed that *SH2B1* knock-out mice develop obesity and hyperglycemia, hyperinsulinemia, glucose intolerance and insulin resistance due to the central role of *SH2B1* in the regulation of glucose and lipid metabolism [39,42,43,44].

In relation to obesity, not only dietary intake and lifestyle, but also genetics have an impact on adiposity being involved in 25–70% in body weight variability [45]. Numerous genes and less frequent variants have been associated with the regulation of energy metabolism [18,39]. Investigations into gene–environment relationships have reported that genes related to nutrient metabolism and transport have a direct association with the requirements of specific nutrients [46]. For example, studies carried out in the Caucasian obese population, following up a hypocaloric diet, showed an association between obese genes and body weight loss, as well as changes in fasting insulin levels and HOMA-IR [47,48]. Furthermore, genetic interactions with environmental factors have been demonstrated to modulate the different responses to a dietary intervention [49,50]. Focusing on NAFLD, there is an increased interest on the study of possible interactions between nutritional factors and genes. An analysis on the Framingham Heart Study reported that increasing diet quality was associated with an improvement of the hepatic health, which is especially of benefit to individuals with a high genetic risk of NAFLD [51]. Moreover, it has been reported that there is a higher effect of I148M *PNPLA3* on steatosis severity in individuals consuming diets poor in vegetables [52]. Concerning the *SH2B* gene, researchers point out that it may be a key target in the regulation of energy balance and body weight [25,53]. A study in mice [39] revealed that neuronal *SH2B1* regulates body weight and nutrient metabolism, this being a genetic variant implicated in glucose and lipid metabolism [25,39]. Another study in children [54] was able to evidence the increased risk (>90%) of developing celiac disease by the genotype of five candidate genes (*SH2B3, RGS1, TAGAP, cREL*, and *LPP*). Together, these findings can help to specifically establish personalized nutritional guidelines that complement the management of NAFLD [21,34]. In our results, dietary and lifestyle characteristics were also evaluated with a semiquantitative FFQ due to the reported association with NAFLD [34]. No significant differences were found in the majority of the nutrients. Besides, similar adherence to the MedDiet score was observed. However, risk allele genotypes referred to higher protein consumption and less MUFA and fiber intake. The higher percentage of dietary fiber could be associated with the lower liver damage presented in non-risk genotypes. This association has also been described previously in NAFLD subjects [3,9]. Second, even though no significant interaction was found between dietary variables and genotype, mechanisms underlying the modulation of macronutrient intake on the *SH2B1* genetic variant are not fully understood and further experimental studies are needed. Lastly, these results have been analyzed at baseline and at one particular moment, so further studies analyzing long-term dietary, lifestyle characteristics and possible gene interactions could be of interest.

Moreover, obesity and insulin resistance are risk factors for NAFLD, but liver biopsy remains the gold standard for the diagnosis [55]. However, it is a flawed and invasive procedure, which can lead to complications [56,57]. Effective screening is essential due to the high prevalence of NAFLD. There is an urgent need to develop a non-invasive and affordable method. In this sense, the study of the relationship of genetics and risk stratification of NAFLD could be useful to provide personalized information about the stage of the disease.

In this article, the *SH2B1* genetic variant was associated with the highest values of the FLI [28]. FLI is a tool for screening liver fat since it is a non-invasive method, inexpensive and is widely available and validated against ultrasonography, but not for diagnosis of NAFLD due to some limitations [58,59]. At the same time, rs7359397_*SH2B1* was associated with the steatosis degree, evidencing a higher risk of higher liver fat accumulation in subjects carrying the risk allele. In this context, Sheng et al. [22] showed that the deletion of *SH2B1* in peripheral tissues promoted hepatic steatosis with an accumulation of liver fat. The results of another article [24] showed defects in the *SH2B1* genetic variant in obese patients with diabetic problems, which are important risk factors for the development of steatosis. When FLI and ultrasonography logistic regression models were adjusted, their association with the polymorphism remained significant, evidencing the influence of the genetic variant in the development of severe stadiums of NAFLD.

On the other hand, one of the main findings of this research was the association between the genetic variant with the content of liver fat measured by MRI. Risk allele participants showed a higher risk for excessive liver fat deposition. Recent studies have indicated that higher levels of abdominal fat, particularly visceral fat, are closely related to NAFLD [60]. In addition, higher visceral fat content was observed in the risk allele group as compared with the other. However, a significant association between visceral fat and *SH2B1* genetic variants were not found in this cohort (*p* = 0.129).

Another relevant result concerned the lipidomic test, which has been used in liver examinations [61]. The OW-Liver Test^®^ and *SH2B1* genetic variants were associated, showing that participants carrying the risk allele presented an increased risk for developing NASH than non-carriers. Therefore, carrying the polymorphism evidenced an important influence on the progression of the disease. This test is a valid, precise and non-invasive method. It has been previously related to more adverse liver markers and general metabolic status in subjects with higher liver damage, also revealing a higher association with steatosis gradation (ultrasonography), as described elsewhere [12]. However, it should be mentioned that even though it is a promising test, it is necessary to develop affordable non-invasive methods in clinical practice.

It is important to mention that most of the models increased their predictive value when adjusting for confounders (ALT, physical activity and diet characteristics). These results may indicate that lifestyle characteristics as well as sex and age should be specifically considered in the personalized management of NAFLD [62,63], along with the traditionally contemplated risk factors such as obesity or insulin resistance [2,13]. Furthermore, hepatic alterations may also be considered. In our case, the prediction of the models was higher when including ALT, but not AST. According to the scientific bibliography, ALT seems to be a sensitive and accurate biomarker of NAFLD [12,64,65], even though in many other cases no associations have been found [12,66]. However, in our analysis, the association of the genetic variant and NAFLD was independent of the adjusting variables, and it is important to note that the prediction of the model was considerably raised when including this genetic variant. Therefore, these findings suggest that NAFLD is associated with this genetic obesity-related polymorphism.

Some limitations concerning this research should be mentioned: Firstly, due to the cross-sectional design of the study, causal inferences cannot be made. Secondly, liver biopsy results were not available to corroborate the precise diagnosis of patients, thus all the associations observed are related to the non-invasive markers of steatosis. Moreover, these data need to be confirmed by histology in future studies [67]. Thirdly, the screening of the participants, including information about competing causes of liver disease, was based on a clinical interview. Additionally, dietary evaluations were carried out using self-reported information of the participants. Thus, subjective measures could produce some biases. The relatively small sample size and the absence of a control group are other limitations. However, the participants included were all well-characterized by the evaluation of NAFLD using recognized techniques such as the MRI and the validated lipidomic test (OWL^®^). Likewise, the combination of TT and CT genotypes could have influenced the results. Despite this, analyses were repeated excluding TT participants and no differences were observed in the results (data not shown).

To our knowledge, a strong point of the study is the potential role of a *SH2B1* genetic variant in the promotion of NAFLD according to a nutritional assessment. Based on the findings reported here, further studies should contemplate the possibility of a risk stratification in accordance with the *SH2B1* genotyping once increasing the proportion of homozygous risk allele carriers in the study population in order to better assess the role of the polymorphism in NAFLD subjects. Moreover, the participants included presented early states of liver damage. Increasing the sample size and enrolling more advanced stages of the disease will also reveal interesting results regarding liver stiffness.

## 5. Conclusions

Carriers of the minor allele of *SH2B1* genotypes have been associated with a higher risk of developing NASH in overweight and obese individuals. These results support the importance of considering genetic predisposition in combination with a healthy dietary pattern in the personalized evaluation and management of NAFLD.

Little is known about the frequency of this risk allele in the general population, the non-obese population, and the non-NAFLD population. Besides, since multiple factors are involved in the pathogenesis of NAFLD, the likelihood of it developing in individuals without the variant should also be studied for the precision management of this disease. Future investigations regarding these issues as well as the possible influence of this genetic variant in the overweight or lean population with NAFLD could be of interest.

## Figures and Tables

**Figure 1 nutrients-12-01260-f001:**
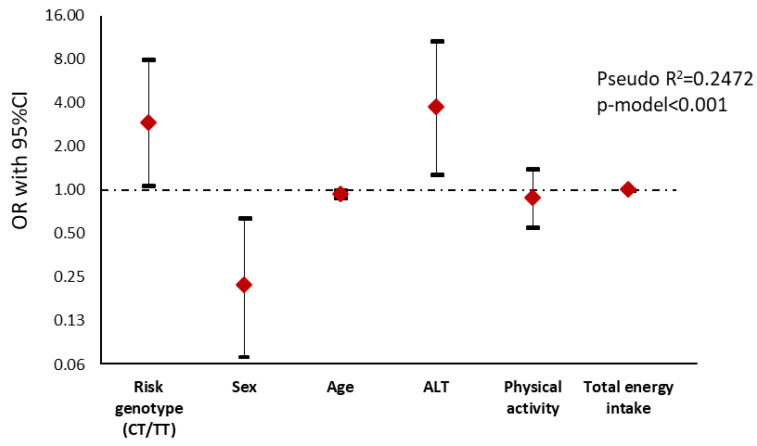
Graphical display of the Odds Ratio (OR) (95% confidence interval) of the logistic regression analysis between Fatty Liver Index and genotype in NAFLD subjects. Fatty Liver Index is the dependent variable and was dichotomized according to the median (0 = FLI < 80 vs. 1 = FLI ≥ 80). Notice that the *y*-axis is on a log scale. ALT, Alanine Aminotransferase.

**Figure 2 nutrients-12-01260-f002:**
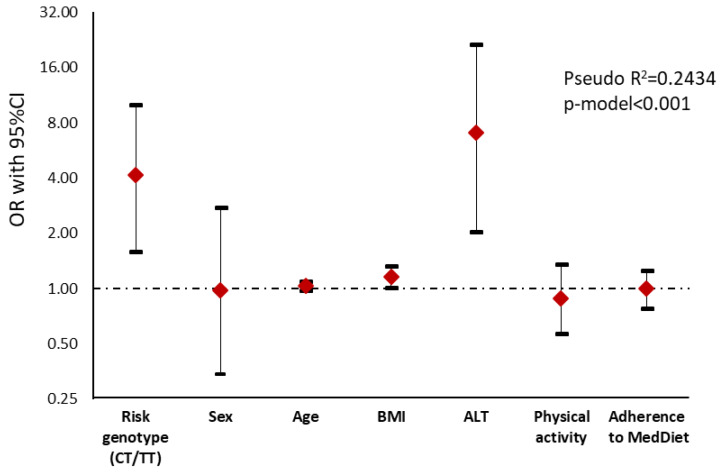
Graphical display of the Odds Ratio (OR) (95% confidence interval) of the logistic regression analysis showing the association between steatosis degree measured by ultrasonography (0 = no or mild steatosis vs. 1 = moderate and severe steatosis) and genotype risk in NAFLD subjects. Notice that the *y*-axis is on a log scale. ALT, Alanine Aminotransferase; BMI, Body Mass Index.

**Figure 3 nutrients-12-01260-f003:**
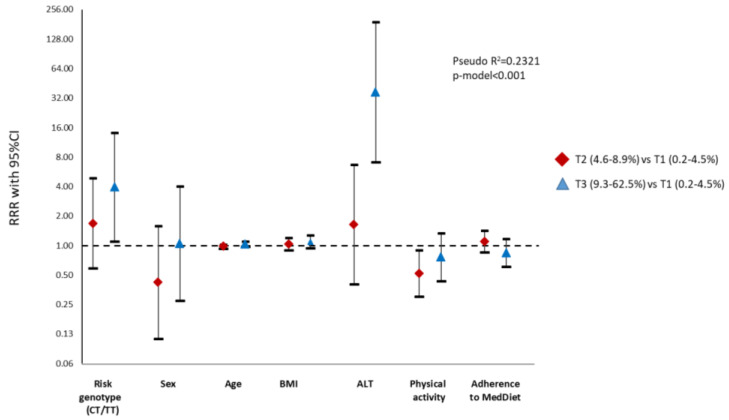
Graphical display of the Relative Risk Ratio (RRR) (95% confidence interval) of the multinomial logistic regression model with *SH2B1* genotype as independent variable and liver fat content (tertiles) assessed by Magnetic Resonance Imaging (MRI) as a dependent variable. Notice that the *y*-axis is on a log scale. ALT, Alanine Aminotransferase; BMI, Body Mass Index.

**Figure 4 nutrients-12-01260-f004:**
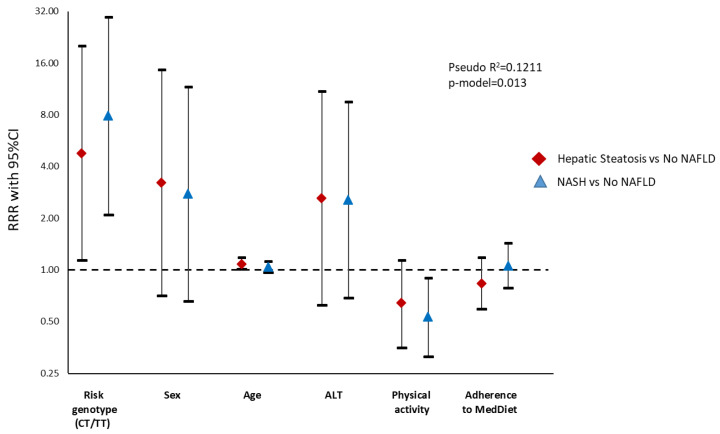
Graphical display of the Relative Risk Ratio (RRR) (95% confidence interval) of the multinomial logistic regression model with *SH2B1* genetic variant as an independent variable and the diagnostic of steatosis or steatohepatitis (by lipidomic analysis) as a dependent variable. Notice that the *y*-axis is on a log scale. ALT, Alanine Aminotransferase; NAFLD: Non-alcoholic Fatty Liver Disease; NASH, Non-alcoholic Steatohepatitis.

**Table 1 nutrients-12-01260-t001:** Body composition and general biochemical parameters of participants according to genotype (risk and non-risk alleles).

	rs7359397_*SH2B1*		
	CC (No-Risk Genotype) n = 54	CT/TT (Risk Genotype) n = 56	*p*-Value
**Body composition**			
Weight (kg)	94.2 (14.6)	97.3 (1.8)	0.133
BMI (kg/m2)	33.4 (4.1)	34.3 (3.6)	0.105
Age (y)	51 (47.0–57.0)	49.5 (45.0–56.5)	0.797
Sex n (%)			
Male	25 (46.3)	37 (66.1)	0.037
Female	29 (53.7)	19 (33.9)	
WC (cm)	108.3 (9.7)	111.1 (9.1)	0.119
DXA Body Fat Mass (kg)	38.0 (32.8–44.6)	38.6 (34.3–44.5)	0.935
DXA Lean Mass (kg)	51.9 (43.9–56.9)	55.5 (49.4–61.8)	0.059
DXA VAT (kg)	2.1 (1.4–3.0)	2.4 (1.8–3.1)	0.129
**Biochemical parameters**			
Glucose (mg/dL)	100 (91.0–111.0)	102.5 (92.5–102.5)	0.421
Insulin (U/mL)	14.1 (9.0–19.8)	20.1 (13.4–25.5)	0.002
TG (mg/dL)	106.5 (76.0–157.0)	127.5 (84.5–160.0)	0.066
TC (mg/dL)	197.5 (40.4)	191.3 (36.8)	0.346
HDL-c (mg/dL)	54.5 (47.0–64.0)	47.0 (40.0–55.5)	0.003
LDL-c (mg/dL)	117.1 (33.9)	115.3 (35.1)	0.788
HOMA-IR	3.5 (2.2–4.7)	4.9 (3.5–6.7)	0.001
HbA1C (%)	5.6 (5.4–5.9)	5.6 (5.4–5.9)	0.622
TyG index	8.5 (8.1–8.9)	8.7 (8.3–9.0)	0.051
Triglycerides/HDL-c (ratio)	1.8 (1.2–3.2)	2.7 (1.7–3.3)	0.021
Waist*TyG index	929.8 (862.3–1001.3)	971.2 (900.8–1033.8)	0.030
HCY (µmol/L)	14.5 (12.2–16.4)	15.1 (11.6–18.1)	0.627
AIP	0.5 (0.2–1.1)	0.9 (0.5–1.2)	0.012

Variables are shown as mean (SD) or as median (IQR) according to their distribution. Categorical variables are presented as absolute (n) and relative frequencies (%). Unpaired *t*-tests and Wilcoxon-Mann-Whitney were used. AIP, Atherogenic Index of Plasma; BMI, Body Mass Index; DXA, Dual-Energy x-ray Absorptiometry; HCY, Homocysteine; HbA1C, Hemoglobin A1c; HDL-c, High Density Lipoprotein-Cholesterol; HOMA-IR, Homeostasis Model Assessment Insulin Resistance; LDL-c, Low-Density Lipoprotein Cholesterol; TC, Total Cholesterol; TG, Triglycerides; TyG index, Triglycerides and Glucose Index; VAT, Visceral Adipose Tissue; WC, Waist Circumference.

**Table 2 nutrients-12-01260-t002:** Daily nutrient intake and lifestyle factors of participants.

	rs7359397_*SH2B1*		
	CC (No-Risk Genotype) n = 54	CT/TT (Risk Genotype) n = 56	*p*-Value
**Energy and macronutrients**			
Total energy (kcal/day)	2649.7 (2181.9–3257.9)	2369.4 (1952.7–2827.7)	0.101
Carbohydrates (%E)	42.3 (6.8)	43.0 (6.8)	0.612
Proteins (%E)	15.7 (14.9–18.1)	17.3 (15.4–20.4)	0.028
Fats (%E)	38.1 (6.14)	36.7 (7.3)	0.308
MUFA (%E)	18.6 (15.5–20.9)	16.2 (14.0–20.0)	0.045
PUFA (%E)	5.4 (4.4–6.7)	5.1 (4.4–6.4)	0.466
SFA (%E)	10.4 (9.3–12.1)	10.3 (9.21–1.8)	0.750
Dietary fiber (g/day)	25.2 (21.2–30.1)	21.3 (17.0–26.8)	0.049
Glycemic Index	53.3 (48.5–58.9)	54.9 (49.1–57.8)	0.988
Glycemic Load	158.6 (98.2–205.6)	139.1 (95.4–176.9)	0.449
**Lifestyle factors**			
Adherence to MedDiet	5.9 (2.1)	6.0 (2.0)	0.678
Physical activity n (%)			
Sedentary	21 (38.8)	24 (42.8)	
Mild	16 (29.6)	13 (23.2)	0.685
Moderated	9 (16.6)	13 (23.2)	
Elevated	8 (14.8)	6 (10.71)	

Variables are shown as mean (SD) or as median (IQR) according to their distribution. Unpaired *t*-tests were carried out. *p* value from paired *t*-test or from Wilcoxon-Mann-Whitney test. Categorical variables are presented as absolute (n) and relative frequencies (%). %E, Percentage of Energy; MUFA, Mono-Unsaturated Fatty Acid; PUFA, Poly-Unsaturated Fatty Acid; SFA, Saturated Fatty Acid.

**Table 3 nutrients-12-01260-t003:** Liver status differences depending on the genotype.

	rs7359397_*SH2B1*		
	CC (No-Risk Genotype) n = 54	CT/TT (Risk Genotype) n = 56	*p*-Value
**Liver markers**			
CRP (mg/dL)	0.2 (0.1–0.4)	0.2 (0.1–0.5)	0.959
FGF21 (pg/mL)	182.0 (96.6–302.0)	214.0 (122.0–478.0)	0.109
AST (U/L)	23.5 (18.0–28.0)	21.0 (18.0–29.0)	0.995
ALT (U/L)	26.0 (18.0–39.0)	30.0 (22.0–46.0)	0.266
Ratio AST/ALT	0.8 (0.6–1.0)	0.7 (0.6–0.9)	0.149
GGT (U/L)	26.0 (19.0–40.0)	32.0 (22.5–44.0)	0.109
FLI	79.8 (66.3–91.2)	87.5 (76.6–93.7)	0.032
**Liver imaging techniques**			
Grade of steatosis (ultrasonography) n (%)			
Mild steatosis	39 (72.2)	22 (39.29)	
Moderate steatosis	11 (20.37)	26 (46.43)	0.001
Severe steatosis	4 (7.41)	8 (14.29)	
TE liver stiffness (kPa)	4.5 (3.8–6.1)	4.5 (3.8–5.6)	0.738
MRI Hepatic Volume (mL)	1701.0 (1409.0–1998.0)	1843.0 (1589.0–2111.0)	0.150
MRI Liver fat—Dixon (%)	4.5 (2.9–8.9)	6.9 (4.4–12.4)	0.055
MRI Hepatic Iron—Dixon (%)	31.8 (28.2–44.2)	32.4 (29.2–38.0)	0.950
**Lipidomic analysis (OWLiver^®^-test) n (%)**			
No NAFLD	16 (29.6)	4 (7.3)	
Hepatic steatosis	14 (25.9)	13 (23.6)	0.006
NASH	24 (44.4)	38 (69.1)	

All variables are shown as median (IQR). Unpaired *t*-test was carried out. *p* value from Wilcoxon-Mann-Whitney test. Categorical variables are presented as absolute (n) and relative frequencies (%). ALT, Alanine Aminotransferase; AST, Aspartate Aminotransferase; CRP, C-Reactive Protein; FLI, Fatty Liver Index; FGF21, Fibroblast Growth Factor 21; GGT, Gamma-Glutamyl Transferase; NAFLD, Non-Alcoholic Fatty Liver Disease; NASH, Non-Alcoholic Steatohepatitis; MRI, Magnetic Resonance Imaging; TE, Transient Elastography.

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
