# Peer review of "Association of the SH2B1 rs7359397 Gene Polymorphism with Steatosis Severity in Subjects with Obesity and Non-Alcoholic Fatty Liver Disease"

_nutrients, 2020, doi:10.3390/nu12051260_

Round 1

Reviewer 1 Report

Dear Editor,

the authors highlight the association between SH2B1 rs7359397 polymorphism and steatosis in a obese NAFLD population, using non-invasive markers of steatosis. This association is interesting. The lack of histology data in my opinion a major concern, and this data should be confirmed with histology data. The authors mention in the results that the lack of histology data is a limitation of the study, but they should add more firmly that the these data need to be confirmed by histology and that the associations that they find are related to the non-invasive markers of steatosis.

In the methods section, the authors still refer to the 4-point ultrasound scale associated to a specific range of % of steatosis. They sould better refer to "normal, mild steatosis, moderate steatosis, or severe steatosis", given that the sonographic evaluation of fatty liver is based mainly on the subjective impression of hepatic echogenicity and posterior attenuation of the ultrasound beam (Strauss et al, 2007; Lee et al 2014), as referred to also in the paper quoted by the authors (Cantero et al, 2019).  

Reviewer 2 Report

On this study the authors aimed to demonstrate that rs7359397 T-allele carriers have a higher risk of developing severe NAFLD compared with non-carriers. This is an interesting study. I have a few minor comments.

  • the authors should show the main results (the association between the gene variant and the liver phenotypes) as figures.
  • Did the authors test for gene nutrients interaction?
  • Is there any interaction between the genotype and the dietary sugars.
  • Was BMI used as a covariate during the analyses?
  • BMI needs to be included in table 7.

Reviewer 3 Report

In this paper the researcher focused their attention on the possible role exerted by the gene SH2B1, and its particular rs7359397 T-allele, in the promotion of more advanced stages of NAFLD. They showed, in a large NAFLD study population, the relationship between the presence of the T allele (homozygosis + heterozygosis = risk group) and the degree of liver steatosis/worsening of metabolic parameters, found a positive correlation also confirmed by logistic regression analyses. The risk group also showed higher risk to develop NASH (assessed by noninvasive method) in comparison to the “non-risk group”.

A strong point of the study, in my opinion, was the analysis of the potential role of SH2B1 gene in the promotion of NAFLD according to a nutritional assessment, that currently represents, the news in this specific field. 

Minor revisions

  • Introduction section, line 100: “… and diet 100 regarding genotypes (risk and no-risk) could have a key role the pathogenesis and dietary 101 management of the disease”. I suggest to reformulate the sentence. It is not completely clear.
  • Statistical analysis section: please specify the methodology used for the sample size calculation.
  • Add future perspectives section at the end of the discussion. In this section the authors could also describe the possibility of a risk stratification in accordance to the SH2B1 genotyping once increased the proportion of “TT patients” in the study population.
  • The fibrosis is the most important prognostic factor in this type of patients. I noticed that there were not significant differences between the two study groups regarding the stiffness. So, I also noticed that the mean and SD stiffness of the enrolled patients is too low. Probably increasing the sample size, enrolling also more advanced stages of the disease this comparison could reveal other interesting results. Please add this point in the future perspectives.

Major revisions

  • Even if the OWLiver® test was used to discriminate the steatosis from NASH, the lack of a histologic evaluation represents the greatest limitation of the study. I suggest to add it in the limitation of the study at the end of discussion section. It is true that the authors describe in the discussion section textually: “Moreover, obesity and insulin resistance are risk factors for NAFLD, but liver biopsy remains the gold standard for the diagnosis [53]. However, it is a flawed and invasive procedure, which can lead to complications [54,55]. Effective screening is essential due to the high prevalence of NAFLD. There is an urgent need to develop a non-invasive and affordable method. In this sense, the study of the relationship of genetics and risk stratification of NAFLD could be useful to provide personalized information about the stage of the disease.”  and following: “However, the design of the current trial is based on validated, non-invasive, and affordable markers, which makes them an optimal form of diagnosis in clinical practice [61,65].”, but in my opinion, it is not enough complete to elucidate the limitation of the study given by this point. It is certainly true that the necessity of some non-invasive diagnostic procedures still remains a big challenge in this field, but for the endpoints proposed by this study the histologic picture of the enrolled patients would have been quite important.
  • I don’t understand the reason why the authors assessed the degree of steatosis by ultrasonography if they used also MRI. The ultrasonography degree of steatosis is not a standardized nor sensible method of comparison.

Round 2

Reviewer 3 Report

I'm satisfied for the authors response.

This manuscript is a resubmission of an earlier submission. The following is a list of the peer review reports and author responses from that submission.

Round 1

Reviewer 1 Report

Hypothesis:

No specific hypothesis is mentioned (until the discussion), they are solely looking for associations. Line 83 – “the aim of the present study was to analyze the association between SHB1 rs7359397 genetic variant and the risk of an advanced stage of NAFLD in obese subjects.”

Rationale:

The rationale for choosing the polymorphism contradicts their introduction where the authors make it sound like this polymorphism has not been studied with liver diseases yet. There is no specific mention of rs7359397 before the aim of the study in the last paragraph of the introduction.

Line 160-161 “the premise for analyzing this specific polymorphism was its association with the steatosis degree assessed by ultrasonography.” The authors might consider elaborating on this statement, including a citation for it, and placing it in their introduction, not the methods section.

General Measurements:

Line – 113-114 “Fatty Liver Index (FLI), which has been validated in a large group of subjects with or without liver disease and has an accuracy of 0.84 (95% CI) in detecting fatty liver.” It is not clear what the authors mean by an accuracy of 0.84, is this 84%? Additionally, what do they mean by (95% CI) there is no interval. Since this is a clinical test, should they mention the false positive and false negative rate?

Line 116 - The FLI <30 is no liver disease and >60 is a marker of liver disease. However, there is no discussion as to whether scores above 60 can be broken into different severities of liver disease. Additionally, there is no source supplied for the statement in line 116. This is important because they compare >80 to <=80, but are these values of FLI clinically distinct or is it just because this is the median?

Statistics:

I take issue with their limited methods used to assess normality. The authors only mention utilizing the shapiro-wilk test; however, it is recommended that this test is paired with the kolgomorov smirnov test. Additionally, they make no mention of visualizing the data through histograms and/or boxplots to assess normality. Additionally, the authors make no mention of assessing for outliers in the data.

There is no issue with the use of logistic regression in the analysis, however, the way the authors frame their methods, it sounds like the authors didn’t arrange a priori the design of their statistical models. It leaves one to wonder if they arranged steatosis degree, liver fat accumulation, and FLI into specific groups in order to uncover significant effects. The authors might consider stating if they planned a priori on dividing the predictors into categories.

The authors mention confounders that they included in the model, but there are no sources for their rationale for including the variables as confounders, additionally, they do not state that they decided on these confounders a priori (a rationale for inclusion of confounders is important, why ALT and not AST or AST/ALT…).

Results:

Line 195 – 197 The authors state that they combined those heterozygous and homozygous for the risk allele (CT and TT) together due to the small sample size. The authors might consider mentioning what limitation arises by combining the two groups.

Table 1. There is a concern that there are significantly more females in the CC (no-risk genotype group compared to the CT/TT group. This could affect some variables such as HDL-c (women tend to have higher levels) and Waist*TyG index, since women tend to have smaller waists. This could also affect Triglycerides/HDL-c, which was found to be significantly different between groups, and total cholesterol. But the authors make limited mention of this fact, especially in the discussion, where they mention the possible interaction with DXA lean mass, but not with HDL-c and Waist*TyG-index.

Table 1 – LDL-c is vastly different between CC and CT, with a similar SD, why is the p-value so large.

Table 1 – Only the percentages of males and females in each group is shown, should show number in each group and then (%) to be consistent with the figure description.

Line 201 – “No significant differences were observed between groups in body composition variables. A marginal significant difference was observed in fat free mass. However, when the analysis was repeated considering men and women separately, no differences were observed between carriers and non-carriers.” Was the analysis only separated for men and women looking at body composition variables? Was there no separate analysis based on gender for the biochemical parameters? If not, the authors might consider running this analysis, because if the findings for the separate genders agree with the findings for both combined, then the distribution of men and women in each group is less of an issue. However, if they the results do not agree, then the ratio of men and women in each group must be addressed.

Table 2. The p values they state in the description of table 2 regarding fiber are different than what is in the table itself (p=0.049 vs 0.051). Physical activity is divided into 4 levels, were unpaired t-tests still used, or was an ANOVA used, because they give one p-value for the whole grouping.

Table 3. Did the authors use ANOVA (or non-parametric counterpart) for group wise comparisons for the Lipidomic analysis and Grade of steatosis? I would recommend the authors expand to include where the significant differences are, with just one p-value it is not clear.

In tables 4, 5, 6, and 7, they fail to bold significant p-values from covariates, not sure if the authors meant to leaves them like that.

Discussion:

Line – 360 – 362 “However, the aforementioned variants are not separately enough themselves to identify patients at risk of developing severe stadiums of NAFLD.” It is not clear what the authors mean with this statement.

Line 371 – 374 they specifically mention their hypothesis. This should be the same as the hypothesis they mention in the beginning. The authors might consider moving this hypothesis statement to the last paragraph of the introduction.

Line 396-397 – “Effective screening is essential due to the high prevalence of NAFLD. There is an urgent need to develop a non-invasive method.” However, the authors go on to say in line 420 about the OW-Liver Test “This test is a valid, precise and non-invasive method.” The authors might be contradicting themselves, unless this test is valid for testing a different liver disease, which the authors might consider mentioning.

Line 408 could be cleared up because it is not clear what logistic regression models they are referring to (FLI and ultrasonography?). The sentence also ends with “influence of the genetic variant on the NAFLD.”

Line 412 – 413 – “Likewise, higher visceral fat content was observed in the risk allele group as compared with the other.” This is not the case when looking at Table 1. DXA VAT, the values are not significantly different. The authors should state what data they are referring too.

Line 428 – “along with the traditionally contemplated risk factors.” Should there be a source for this statement?

Line 443-444 “besides, this genetic variant has been scarcely studied in association with NAFLD.” This seems to contradict earlier statements by the authors. There is no mention in the introduction of studies that relate this variant to NAFLD. I recommend the authors are more specific with their languages, additionally, the authors need to provide citations for this statement.

Conclusion:

The authors might consider shortening the discussion and lengthening the conclusion. To do this a few of the paragraphs in the discussion could be used in the conclusion.

Contribution to the field:

This article provides strong evidence that the SHB1-rs7359397 genetic variant is strongly associated with a more severe level of NAFLD in an obese population. However, there is already such a high prevalence of NAFLD in obese populations that this genetic predisposition might not be as important. Especially since both groups, based on the FLI had a fatty liver (FLI >60) and at least some level of steatosis as measured by ultrasonography. However, it is an important finding because identifying multiple genetic predispositions can aid in the development of genetic testing for the prevention of these types of chronic diseases. The authors might consider recommending future directions for research in regard to this genetic variant and NAFLD, such as looking at a normal or overweight population instead of obese individuals.

Reviewer 2 Report

I read with great interst the paper from del Campo et al. The topic is of interest. The paper suggests an association between the SH2B1 rs7359397 variant and the risk of NAFLD in obese subjects.

There are, however, different concerns about of this study.

  1. The author should describe better the genetic variants they study and how the function of these genes would fit in the onset of NAFLD. Since the SH2B1 is altered in obesity, could the association found in the paper due to the metabolic impairment of the patients? The authors should clarify better this point.
  2. The authors performed genotype screen of epithelial buccal cells and did not use blood samples. The use of buccal test is questionable and it is not widely used in NASH. The authors should provide evidence that the presented results are robust.
  3. The authors merged the patients with CT and TT variants because of lack of balanced groups. This choice might have influenced the results.
  4. The characterization of the patients has some limitation. The authors use the HOMA IR and HbA1C but no OGTT was performed. Moreover one crucial limitation of this paper is the lack of liver histology. OWLiver test a non-invasive metabolomic based test, which is promising but needs further validation in independent cohorts.
  5. The authors should clarify why they chose the Fibroscan cut off of 7 and 12 for respectively fibrosis (any fibrosis? Significant fibrosis?) and cirrhosis.
  6. The authors should revise the ultrasound classification of liver steatosis. E.g. according the classification by Saverymuttu et al., Br MedJ 1986; 292:13–15, which is based on the image characteristics (echogenicity of the liver parenchyma compared to the right kidney, the aspect of the portal vein brunches wall and the echo-attenuation in the deep planes). I wonder how the investigators are able to quantificate the percentage of liver fat based on ultrasound images.

Round 2

Reviewer 1 Report

I believe that the authors have sufficiently responded to the previously mentioned concerns and questions.

Does the introduction provide sufficient background and include all relevant references?

The authors discuss in detail NAFLD and develop a rationale for why investigating genetic susceptibility is important for identifying those with higher risks of developing advanced stages of NAFLD. The introduction was a bit lacking in information about why the specific genetic variant (rs7359397) of SH2B was chosen, but it appears that the authors have added additional information to this portion of the introduction. Additionally, the introduction section includes an adequate number of relevant sources that support the information they provide.

Is the research design appropriate?

The research design that the authors chose is appropriate to answer the specific hypothesis that the researchers were trying to test. This research design is especially useful for clinical environments, which the results are relevant to. Even though there are limitations to the design that the authors used, they did a good job in addressing them in the research design (inclusion of confounders in logistic regressions), and they made a point to mention the limitations in the discussion.

Are the methods adequately described?

The methods that the authors used are describe in adequate detail. The authors do a good job breaking the methods section up into distinct sections with relevant citations for each of their statements. The authors also have added to their statistics section in certain important areas that were raised (normality, outliers, and selection of confounders).

Are the results clearly presented?

The use of tables by the authors helps to clearly show the results of the study. The authors also to a good job of describing the results and pointing out potential problems such as the unequal distribution of males and females in the different genotype groups. There were some errors in the display of some of the tables in the previous draft, but the authors appear to have fixed these issues. The authors do a good job of describing the logistic regression models that they used for the study. The interpretation of the odds ratios is straightforward, but the authors might want to consider elaborating on the last two logistic regressions (table 6 and 7). Specifically, they could mention what the interpretation of the beta values in lines 318 – 319 (β = 1.36 and β = 2.01) mean in terms of their model.

Are the conclusions supported by the results?

The authors used multiple methods for assessing liver damage, which adds to the strength of their conclusions, since all these methods showed an increased risk for the risk genotype in their models. The results support the conclusion that carriers of the minor allele of SH2B1 genotypes have been associated with a higher risk of developing NASH. The authors might consider narrowing the conclusion by stating that it is associated with a higher risk of NASH in overweight and obese individuals. Other than this small point, the authors do a good job in not overstating their findings, which is appreciated.

Reviewer 2 Report

The authors replied to the questions. The are still concerns about study design. The lack of histology data is a major weakness of the paper. 
